# Transcriptomic Analysis to Unravel Potential Pathways and Genes Involved in Pecan (*Carya illinoinensis*) Resistance to *Pestalotiopsis microspora*

**DOI:** 10.3390/ijms231911621

**Published:** 2022-10-01

**Authors:** Yu Chen, Shijie Zhang, Yuqiang Zhao, Zhenghai Mo, Wu Wang, Cancan Zhu

**Affiliations:** Institute of Botany, Jiangsu Province and Chinese Academy of Sciences, Nanjing 210014, China

**Keywords:** pecan, RNA-Seq, fruit black spot, *Pestalotiopsis microspora*, WRKY TFs, transcriptomics

## Abstract

Fruit black spot (FBS), a fungal disease of pecan (*Carya illinoinensis* (Wangenh) K. Koch) caused by the pathogen *Pestalotiopsis microspora*, is a serious disease and poses a critical threat to pecan yield and quality. However, the details of pecan responses to FBS infection at the transcriptional level remain to be elucidated. In present study, we used RNA-Seq to analyze differential gene expression in three pecan cultivars with varied resistance to FBS infection: Xinxuan-4 (X4), Mahan (M), and Wichita (W), which were categorized as having low, mild, and high susceptibility to FBS, respectively. Nine RNA-Seq libraries were constructed, comprising a total of 58.56 Gb of high-quality bases, and 2420, 4380, and 8754 differentially expressed genes (DEGs) with |log2Fold change| ≥ 1 and *p*-value < 0.05 were identified between M vs. X4, W vs. M, and W vs. X4, respectively. Kyoto Encyclopedia of Genes and Genomes (KEGG) metabolic pathway analyses were performed to further annotate DEGs that were part of specific pathways, which revealed that out of 134 total pathways, MAPK signaling pathway, plant–pathogen interaction, and plant hormone signal transduction were highly enriched. Transcriptomic profiling analysis revealed that 1681 pathogen-related genes (PRGs), including 24 genes encoding WRKY transcription factors, potentially participate in the process of defense against *Pestalotiopsis microspora* infection in pecan. The correlation of WRKY TFs and PRGs was also performed to reveal the potential interaction networks among disease-resistance/pathogenesis-related genes and WRKY TFs. Expression profiling of nine genes annotated as TIFY, WRKY TF, and disease-resistance protein-related genes was performed using qRT-PCR, and the results were correlated with RNA-Seq data. This study provides valuable information on the molecular basis of pecan–*Pestalotiopsis microspora* interaction mechanisms and offers a repertoire of candidate genes related to pecan fruit response to FBS infection.

## 1. Introduction

Pecan (*Carya illinoinensis*), which belongs to the Juglandaceae family, is a valuable nut crop in the world. It originated in North America and has been widely planted in China in recent years, including in Jiangsu and Anhui Provinces [1]. Fruit black spot (FBS) is a critical pathogen observed in pecan cultivation that impedes plant growth, limits overall crop productivity, and affects seed quality [2]. At present, reports on pecan–*Pestalotiopsis microspora* interaction mechanisms are limited. Profiling the mechanism of pecan defense against FBS infestation is of fundamental biological interest and such knowledge could provide genetic ideas to manipulate the yield and quality of pecan fruits.

The yield and quality of pecan are strongly affected by various biotic and abiotic stresses, such as salt [3,4], drought [5], and pathogens [6]. Among the wound pathogens, black spot is a fungal disease caused by *Pestalotiopsis microspora*, which leads to decayed pecan fruits [2]. In some other plants, such as citrus, this pathogen can produce mechanical pressure against host plant cells and can steal nutrients and water from infected plant tissues, and metabolites including enzymes (cutinase, protease, and cell wall hydrolase) and toxins (peptides, terpenoids, and others proteins) that are harmful to the normal physiological behavior of the host will be released [7,8,9]. Besides the pathogenic ability, the response of the host plant can also be induced [10]. Pattern recognition receptors (PRRs) are the first signal of pathogen recognition [11]; when leaves or fruits are exposed to the *Pestalotiopsis microspore* environment, PRR signals can lead to a series of signal transductions and induce plant defense responses through the recognition of pathogen-associated molecular pattern molecule (PAMP) signatures [12]. In addition to these signals and enzymes, mitogen-activated protein kinase (MAPK) signaling [13], jasmonate (JA) signaling [14] and pathogenesis-related (PR) proteins [15] have also been reported as part of the host response to pathogen infection, but the mechanisms in the pecan–Pestalotiopsis microspora interaction system remain unclear.

Along with the various functional genes, enzymes, and pathogenesis-related proteins, many transcription factor (TF) families have been suggested to play important roles in transcriptional reprogramming associated with plant–pathogen response. For example, bZIP TF was upregulated after *Ustilago maydis* infection, exhibiting a prominent function in maize response to pathogen attack [16]. The overexpression of *OsWRKY89* positively modulated pathogen defense response by elevated lignin accumulation in rice [17]. Other TF families, including ERF [18], NAC [19], and MYB [20], also modulate plant resistance against pathogens. Among these TFs, WRKY was widely reported to be involved in plant pathogen resistance. In walnut, *JrWRKY21* was recently reported to interact with *JrPTI5L* to activate expression of *JrPR5L* and gain resistance to *Colletotrichum gloeosporioides* infection. However, the roles of candidate TFs in pecan–pathogen interactions are not well understood.

Transcriptome sequencing (RNA-Seq) is an effective technology based on high-throughput sequencing used to study metabolic processes, quantify gene expression, and identify key genes associated with traits of interest. In pecan, RNA-Seq analysis provides insights into transcriptome profiles associated with several traits, such as flavonoid biosynthesis in kernels [21], parallel expression patterns between allergens and lipid metabolism [22], and pistillate flowering [23]. In other plants, RNA-Seq studies of the genes and genetic mechanisms of plant–pathogen interactions have been conducted under different conditions, including cucumber *Alternaria* leaf spot infection [24], poplar leaf spot infection [25], and maize response to gray spot disease [26]. However, transcriptome profiling analyses of resistance to FBS are limited. To explore the mechanism of resistance to FBS in pecan, the transcriptional profiles of three pecan cultivars with varied resistance to FBS infection were determined, which can provide a useful resource to aid in the selection of target resistance genes to enhanced FBS control of pecan.

In the present study, we aimed (1) to compare the differentially expressed genes (DEGs) among pecan cultivars in response to FBS infection, (2) to elucidate the possible pathways related to pecan–FBS interactions, and (3) to identify candidate resistance genes and TFs against FBS disease. This study can provide a theoretical basis to understand the mechanisms of woody oil plant–pathogen interactions.

## 2. Results

### 2.1. Evaluation of Pecan Fruits against FBS Infection

The results of FBS incidence rate and index for three pecan cultivars are shown in Figure 1, along with the macroscopic observations of fruits in the three cultivars. The results show that the FBS incidence rate was significantly higher in W than in M and X4. The development pattern of the FBS incidence index in the three pecan cultivars looks similar to the FBS incidence rate, which categorize X4, M, and W as having low, medium, and high susceptibility to FBS infestation, respectively.

### 2.2. Library Construction and Sequencing

Fruit samples of three pecan cultivars were subjected to RNA-Seq. This high-throughput sequencing generated at least 6.09 Gb 150 bp clean bases from each library (Table 1). After stringent filtering, 58.56 GB of clean data from nine libraries was obtained, with average Q30 of 91.29%. Then, clean reads were mapped to the pecan reference genome (https://www.ncbi.nlm.nih.gov/assembly/GCF_018687715.1, accessed on 3 June 2021). The total mapped reads among the nine libraries ranged from 84.11 to 88.06%, with at least >65.05% unique mapped reads, and the sequenced samples were also subjected to principal component analysis (PCA) and Pearson correlation analysis (Figure 2A). Correlation within group was at least R^2^ > 0.82. PCA of the nine libraries (three samples, three biological replicates each; Figure 2B) revealed that the samples from the three cultivars could be clearly distinguished, confirming the existence of varied resistance to FBS infection in these cultivars. PCA data (PCA1 variance explained 33.09% and PCA2 15.24%) represent 48.33% of variance across the two dimensions. All results suggest that the sequencing depth and quality of data were sufficient for further analysis. The present sequencing data were deposited in NCBI SRA and are available under accession number PRJNA824689.

### 2.3. Differentially Expressed Genes (DEGs)

In this study, gene expression levels were estimated via the FPKM method. Differentially expressed genes (DEGs) with |log_2_Fold change| ≥ 1 and *p*-value < 0.05 were considered as significantly different. A total of 1307, 2162, and 4773 genes were upregulated and 1113, 2218, and 3981 genes were downregulated between M vs. X4, W vs. M, and W vs. X4, respectively (Figure 3A,B). According to the results, the majority of DEGs were observed in the comparison W vs. X4, with W and X4 being the most susceptible and resistant to FBS infestation, respectively.

### 2.4. GO and KEGG Functional Enrichment Analysis of DEGs

To explore the functions of detected DEGs, GO-based enrichment annotation was performed, with a threshold value (*p*-value) < 0.05 set to evaluate significantly enriched GO pathways. In the comparison of W vs. X4, a total of 6869 DEGs (78.46%) were annotated as “molecular function”, which contained the majority of GO terms, followed by 6667 DEGs (76.15%) annotated as “biological process”, and 6470 DEGs (73.9%) annotated as “cellular component”. Through GO assignment, the DEGs were divided into 38 functional groups. The major GO subcategories of all transcripts among the three cultivars are shown in Figure 4. The GO term “catalytic activity” (GO:0003824) includes the top two DEGs in the comparison of M vs. X4, W vs. M, and W vs. X4, which suggests that catalytic enzymes might be involved in the pecan–FBS interaction process. In addition, DEG mapping to KEGG pathways was also performed to identify the candidate pathways and genes associated with the pecan–FBS interaction process. The significantly enriched (*p*-value < 0.05) pathways are shown in Table 2. Among the three comparison panels, the plant–pathogen interaction (ko04626), MAPK signaling (ko04016), and plant hormone signal transduction (ko04075) pathways are all significantly enriched (*p*-value < 0.05). The plant–pathogen interaction pathway includes the highest number of DEGs in M vs. X4 and W vs. X4, which shows that the DEGs in this pathway were most activated after FBS infection, and the majority of genes in the pathway were more highly expressed in FBS-resistant variety ‘Xinxuan-4′ (Appendix A).

### 2.5. Pathogen-Related Genes (PRGs)

By analyzing DEGs in the plant–pathogen interaction pathway (Appendix A) and mapping them to the PRG database (http://prgdb.crg.eu/, accessed on 3 June 2021), 1602 PRGs (Appendix A) were identified; homologues of these in other plants were reported to confer resistance to pathogens. All PRGs were grouped into 12 classes (class descriptions are available online: http://prgdb.crg.eu/wiki/Category:Classes, accessed on 3 June 2021) based on the presence of specific domains or partial domains (Figure 5), in which 504 RLP, 287 TNL, and 260 NL class genes were identified, representing the top three classes with the highest numbers of PRGs. The varying resistance capability between pecan and FBS may partly result from differences in the number of PRGs. Comparing the DEGs of plant–pathogen interaction (ko04626) from the KEGG pathway, 25 DEGs (|log_2_Fold change| ≥ 1 and *p*-value < 0.05) annotated as disease-resistance/pathogenesis-related proteins were selected as candidate genes involved in pecan–FBS interactions (Appendix A). Pathogenesis-related protein 1-like gene (*LOC122309141*) showed the highest expression among the 25 DEGs, and the expression (FPKM value) of this gene was 20- and 108-fold higher in M and W, respectively, than X4 (Appendix A).

### 2.6. Transcription Factor Analysis

In the present study, a total of 305 differentially expressed genes (|log_2_Fold change| ≥ 1 and *p*-value < 0.05) annotated as transcription factors (TFs) were identified among the compared DEGs, belonging to 24 major transcription factor families (Appendix A). The DEGs encoding transcription factors were mostly members of the bHLH (36/305), MYB (29/305), NAC (26/305), and WRKY (24/305) families (Figure 6). As shown in the figure, the expression trend of MYB and NAC family members was not consistent, and the highest expression levels of all WRKY TFs were observed in W. DEGs of WRKY families were more highly expressed in varieties susceptible to FBS infection, which was why we constructed the correlation of WRKY and PRGs.

### 2.7. Correlation of Pathogen-Related Genes (PRGs) and WRKY TFs

In order to identify the potential functions of WRKY TFs associated with the expression of PRGs involved in pecan–FBS interactions, a correlation network of gene expression between WRKY TF family members and PRGs was constructed (Figure 7 and Appendix A). From the results, we can observe that specific PRGs were positively/negatively correlated with all TFs; for example, disease-resistance protein RUN1 (*LOC122297372*) and pathogenesis-related protein 1 (*LOC122309453*) were positively correlated with all WRKY members, and disease-resistance proteins At4g27190 (*LOC122307932*) and At1g58602 (*LOC122319395*) were negatively correlated with all WRKY members.

### 2.8. Confirmation of DEGs by qRT-PCR

To validate the expression patterns of RNA-Seq data, the transcript levels of nine candidate PRGs, including three TIFY protein genes, three WRKY TFs, and three disease-resistance protein genes, were determined by qRT-PCR to verify the reliability of the expression profiles of DEGs identified among the nine sequenced pecan samples (three biological repeats per sample). The RNA-Seq results were highly consistent with the trends obtained by qRT-PCR (Figure 8, blue bars vs. black lines). These results verify the reliability and reproducibility of the RNA-Seq data.

## 3. Discussion

FBS is one of the most pervasive and damaging fungal diseases of pecan leaves and fruits [27], yet there are few studies on the molecular response of pecan fruit to this pathogen. In the present study, to determine the molecular basis of pecan fruit response to FBS and to analyze the expression profile changes of disease-resistance genes, we collected an extensive set of RNA-Seq data covering the infection-related transcriptome in pecan fruits of three cultivars that differ in their susceptibility to infection. The results provide new insights into the response of pecan fruit to one of its most damaging pathogens, FBS. The findings suggest that differential regulation of PRG pathways and expression may play a vital role in FBS-induced defense responses. GO and KEGG pathway analysis revealed that genes of catalytic activity (Figure 4), plant–pathogen interaction, MAPK signaling, and hormone signal transduction (Table 2) were enriched after FBS infection. In the present study, there was a cascade of differentially expressed defense response genes, which might lead to an incompatible interaction between host and pathogen, resulting in the synthesis of antimicrobial secondary metabolites that alleviate the symptoms caused by FBS.

Enzyme-catalyzed disease resistance is a commonly observed phenomenon in plants; when fruits or leaves are bruised or wounded, reactions occur [28]. Enzymes involved in resistance against microbial pathogens in plants have been widely reported, such as peroxidase and polyphenol oxidase involved in resistance to *Macrophomina phaseolina* (Tassai) Goid infection in *Brassica juncea* [29]. The role of peroxidases in plant disease resistance processes has also been described in maize [30], tomato [31], and wheat [32]. In addition, other catalytic enzymes have been reported as signaling factors that increase plant biotic stress resistance, such as menthone reductase, which positively regulates pepper defense against a broad range of pathogens [33], and pectic enzymes, which are virulence factors of plant pathogens [34]. In the present study, the most significant enriched terms identified in GO analysis were in ‘catalytic activity’, with 1475, 2837, and 784 genes in the GO pathway detected in the comparisons of W vs. M, W vs. X4, and M vs. X4, respectively (Figure 4). DEGs of the ‘catalytic activity’ pathway may be involved in the interactions of pecan fruit and FBS.

From KEGG enrichment analysis, hormone signal transduction, plant–pathogen interaction, and MAPK signaling pathway were significantly enriched among all compared panels of W, M, and X4 (Table 2). In the comparison of W vs. X4, 489 and 651 DEGs were observed in the MAPK signaling and plant–pathogen interaction pathways, respectively. Around two-thirds of the genes were more highly expressed in W (Table 2), which suggests that more genes in W are activated to alleviate the symptoms caused by FBS. The pathway genes pathogenesis-related protein 1 (*LOC122309453*), threonine-protein kinase GSO2 (*LOC122310044*), and allene oxide cyclase 4 (*LOC122309351*) were only expressed in W (Appendix A). Plant MAPK cascades can play a vital role in defense against pathogen attack. MAPKs are involved in signaling multiple defense responses, including defense hormones, signaling of plant stress, and reactive oxygen generation [35]. The function of MAPK in plant disease resistance has been reported in *Arabidopsis* [36], tomato [37], cotton [38], and rice [39]. MAPK functions in plant defense signaling networks by transducing and amplifying signals generated by NB-LRR or PRRs into defense responses [40]. Candidate DEGs of the MAPK pathway that were identified in our study include LRRs (e.g., *LOC122277177* and *LOC122279408*) and MAPKs (e.g., *LOC122292765* and *LOC122294474*), which could be useful for understanding pecan–FBS interaction networks.

Plant MAPK cascades have also been implicated in the regulation of defense hormone biosynthesis and hormone signal sensing [41,42]. Hormones are important secondary signaling molecules, and the plant hormones ethylene, jasmonate (JA), salicylic acid (SA), and brassinosteroids (BRs) are recognized as important components of plant defense signaling [43,44,45,46,47]. Specific pathogen-/stress-triggered biosynthesis and signal transduction of defense hormones induce the expression of an array of defense-related genes, leading to diverse defense responses of donor plants. In the present study, plant hormone signal transduction was significantly enriched in the KEGG pathway, and 421 genes in this pathway were differentially expressed in W vs. X4 (Table 2 and Appendix A). Among them, JA signaling has frequently been reported in plant–disease interaction networks, and in this study, three genes of the JA signaling pathway encoding TIFY proteins (TIFY 10B, *LOC122280500*; TIFY 9, *LOC122308877* and *LOC122312853*) were highly expressed in W (Appendix A). These results indicate that the JA and other signaling pathways are also involved in FBS defense in pecan.

Apart from the pathway genes, there is also genetic evidence showing that TFs are also involved in plant defense processes [20,48], and the functional category of TFs was significantly enriched in the expression profiles of many biotic defense-related comparative transcriptomics studies [49]. In the present study, 305 TFs belonging to 24 major transcription factor families were differentially expressed after FBS infection, mainly bHLH, NAC, MYB, and WRKY (Figure 6 and Appendix A). Among them, the expression patterns of WRKY TF family members were highly consistent. All 24 members showed the highest expression in W, while much lower expression of WRKYs was observed in M and X4 (Figure 6 and Appendix A). In other studies, *CaWRKY1* [50], *BnWRKYs* [51], *MdWRKY 75d/e* [52], and *JrWRKY21* [53] were reported to play major roles in plant–pathogen/plant–disease resistance. Together with the correlation results of pathogen-related genes (PRGs) and WRKY TFs (Figure 7), we can provisionally conclude that upregulation of WRKY TFs in W implicates their role in the regulation of transcriptional reprogramming and function against FBS infection in pecan fruits.

Pecan FBS resistance is an extremely complex process that is coordinated by multiple factors and involves the participation of many genes and catalytic enzymes. Understanding pathogen–plant interactions is important not only from an evolutionary and ecological perspective but also for the development of novel crop protection strategies (pathogen-resistance breeding). Overall, this study represents a first step toward providing genetic information to understand the molecular mechanisms involved in resistance to FBS, and the transcriptome data generated here will also help in guiding further research to develop new strategies/ideas for disease management in pecan.

## 4. Material and Methods

### 4.1. Plant Material and Sample Collection

Six-year-old pecan (*Carya illinoinensis*) plants of three cultivars, Xinxuan-4 (X4), Mahan (M), and Wichita (W), were planted at Jintan forest farm (31°42′34″ N, 119°22′8″ E), Changzhou, Jiangsu Province, China, under the same growth conditions (humidity, temperature, light, etc.). The resistance to and susceptibility of pecan fruits of different cultivars against FBS infestation were calculated in 2019 [27]. The method of Chen et al. [27] and Jiang et al. [54] was employed to evaluate the FBS incidence rate and severity index of the 3 pecan cultivars. The FBS incidence rate (FBS IR) was calculated as total infested fruits/total fruits, and the FBS severity index (FBS SI) was calculated as Σ(α_i_·n)/(A·N). In the equation, α_i_ is the numerical disease rating (0, 1, 2, 3, 4), n is the number of infested fruits under each disease rating for each plant, A is the maximum disease rating of 4, and N is the total number of infested fruits in each plant. The exocarp of pecan fruits (20 fruits were collected for each replicate) was harvested from plants naturally infected with FBS before maturity at same time during August 2019. Three plants of each cultivar were used in this study, and each plant was considered as a biological replicate. Samples were promptly frozen in liquid nitrogen and stored at −80 °C until further experiments.

### 4.2. RNA Preparation and Transcriptome Sequencing

Total RNA was isolated from the exocarp of pecan fruits with a plant RNA extraction kit (Fuji, China). For this process, 100 mg frozen fruit samples were ground in a mortar under liquid nitrogen, and the resultant powder was used for RNA extraction, referring to Mo et al. [1]. The concentration and integrity of RNA samples were assessed by agarose gel electrophoresis, NanoDrop 2000 spectrophotometer (Thermo Fisher Scientific, Waltham, MA, USA), and Agilent 2100 bioanalyzer (Agilent Technologies Inc., Santa Clara, CA, USA). RNA samples in correct electrophoresis stripes (clearly visible 5S, 18S, and 28S ribosomal RNA bands) with an A260/A280 ratio of 1.8–2.1 were used as template material for qRT-PCR and RNA sequencing. On the basis of the Illumina standard procedure, 9 libraries were prepared (3 biological replicates per sample) and sequenced with the Illumina HiSeq 2500 platform (BGI, Shenzhen, China).

### 4.3. RNA-Seq Data and Differentially Expressed Gene (DEG) Analysis

Raw data from the sequence platform were trimmed to remove low-quality sequences (i.e., adapter sequences and poly-N tail reads). After that process, the data were obtained and mapped to the pecan reference genome (https://www.ncbi.nlm.nih.gov/assembly/GCF_018687715.1, accessed on 3 June 2021) with HISAT2 (v2.0.4) [55]. The expression levels of genes were calculated according to fragments per kilobase per million reads (FPKM) values. Differentially expressed genes (DEGs) were identified by using the DESeq (v1.4.5) package. Thresholds of *p*-value < 0.05 and |log2Fold change| ≥ 1 were set to determine whether genes had significant expression between two samples.

### 4.4. GO and KEGG Enrichment Annotation of DEGs

To gain insight into their main biological functions and pathways, all DEGs were mapped to terms in the Gene Ontology (GO; http://www.geneontology.org/, accessed on 3 June 2021) and Kyoto Encyclopedia of Genes and Genomes (KEGG; https://www.kegg.jp/, accessed on 3 June 2021) databases. GO enrichment was analyzed by AgriGO with *p* < 0.01 and FDR < 0.05 [56] and KEGG enrichment was analyzed by the KOBAS 3.0 tool (http://kobas.cbi.pku.edu.cn/, accessed on 3 June 2021); the significance level of the pathways was corrected using a rigorous threshold (*p*-value < 0.05) via Bonferroni correction.

### 4.5. Screening of Transcription Factors (TFs)

To screen the genes annotated as transcription factors (TFs), we used getorf (EMBOSS:6.5.7.0, http://emboss.sourceforge.net/apps/cvs/emboss/apps/getorf.html, accessed on 3 June 2021) to detect the open reading frames (ORFs) of UniGene. Furthermore, we used hmmsearch (v3.0, http://hmmer.org, accessed on 3 June 2021) to compare ORF sequences with the domains of transcription factor proteins, then identify the UniGene entry according to the transcription factor family characteristics described by plantTFDB (v5.0, http://planttfdb.gao-lab.org/index.php, accessed on 3 June 2021). TFs with absolute |log2Fold change| ≥ 1 and *p*-value < 0.05 across at least one of the three comparisons were considered as significant.

### 4.6. Identification of Pathogen-Related Genes (PRGs)

To identify the pathogen-related genes (PRGs) potentially involved in pecan–Pestalotiopsis microspora interaction, we used Diamond software (v0.8.31; https://github.com/bbuchfink/diamond, accessed on 3 June 2021) to map the DEGs to the Resistance Gene Database (PRGdb, v2.0; http://prgdb.crg.eu/, accessed on 3 June 2021) based on the query coverage and identity requirement.

### 4.7. Correlation Network Analysis of PRGs and WRKY TFs

TFs can regulate the expression of target genes. To explore the potential correlation between PRGs and WRKY TFs identified in the study, correlation coefficients were determined by SPSS software (SPSS Inc., Chicago, IL, USA) and the correlation figures were drawn using OmicStudio tools (https://www.omicstudio.cn/tool, accessed on 3 June 2021).

### 4.8. Candidate DEG Validation by qRT-PCR

Nine DEGs annotated as TIFY, WRKY TFs, and PRGs were selected from RNA-Seq data and analyzed by qRT-PCR to validate the expression patterns, as described previously [57]. Specific primers (Appendix A) for qRT-PCR were designed with Primer Premier 5 software (Palo Alto, San Francisco, CA, USA). The 18S ribosomal RNA (AF174619.1) was selected as internal reference gene in this study [22]. qRT-PCR was conducted using an ABI7500-fast PCR system according to the manufacturer’s protocol (Applied Biosystems, Carlsbad, CA, USA). All reactions and non-template controls were carried out in triplicate. Relative mRNA expression was calculated by the 2^−ΔΔCt^ method [58]. Data are shown as mean ± standard error (SE) (*n* = 3).

### 4.9. Statistical Analysis

Figures were made by GraphPad Prism version 8.0.1 (San Diego, CA, USA). All data analyzed in the current study are expressed as mean ± standard error (SE). SPSS version 17.0 (SPSS Inc., Chicago, IL, USA) and Excel software were used for statistical analysis. One-way analysis of variance with Duncan’s t-test was used to evaluate significant differences (*p* < 0.05). TBtools software (v.1.09876) was used for heatmaps.

## 5. Conclusions

In the present study, we compared the differentially expressed genes (DEGs) among three pecan cultivars in response to FBS infection to develop of a hypothesis regarding the molecular mechanism underlying pecan fruit–FBS interactions. By comparing the transcriptomes among the three pecan cultivars, MAPK signaling, plant–pathogen interaction, and plant hormone signal transduction pathways, as well DEGs in these pathways, were identified as potentially leading to the FBS susceptibility of pecan fruits. A number of regulatory proteins encoding the WRKY TFs were identified and found to play critical roles in the process; however, functional characterization of such genes to delineate their precise roles in FBS resistance is necessary, as these genes may be useful in breeding programs to develop FBS-resistant pecan varieties.

## Figures and Tables

**Figure 1 ijms-23-11621-f001:**
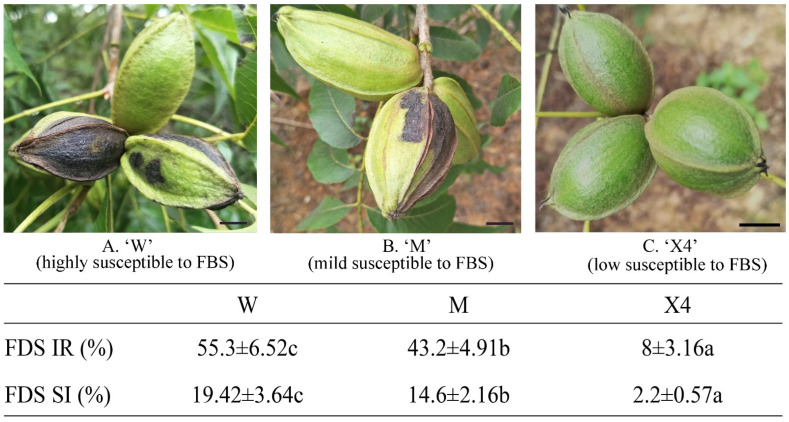
Symptoms and incidence rate (IR)/severity index (SI) of pecan fruits after FBS infection. Percentage of IR/SI is expressed as mean ± standard error (*n* = 12). Scale bar = 1 cm. IR was calculated as total infested fruits/total fruits; SI was calculated as Σ(α_i_·n)/(A·N). X4, Xinxuan-4; M, Mahan; W, Wichita. Different letters (a–c) indicate significant difference (*p* < 0.05).

**Figure 2 ijms-23-11621-f002:**
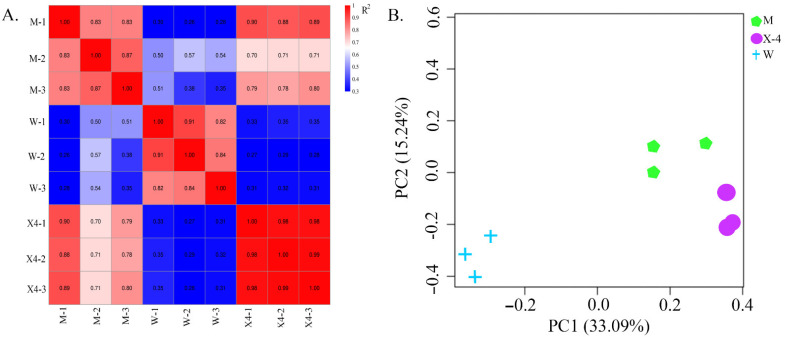
(**A**) Pearson’s correlation matrix and (**B**) principle component analysis of differentially expressed genes (DEGs) to evaluate correlations and variance between samples. Three biological replicates of each sample set were performed. X4, Xinxuan-4; M, Mahan; W, Wichita.

**Figure 3 ijms-23-11621-f003:**
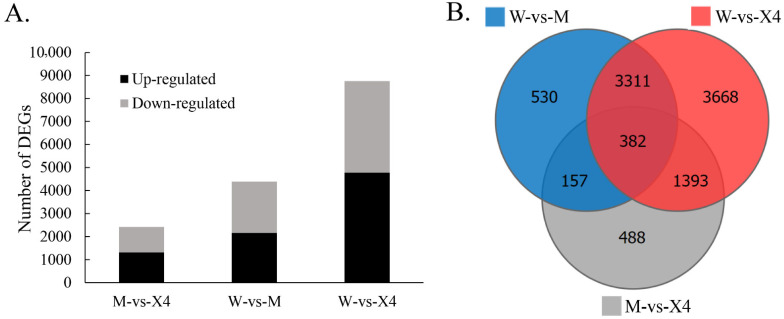
Differential gene expression in pecan fruits in the presence of FBS in resistant/susceptible cultivars. (**A**) Number of upregulated and downregulated DEGs (|log2Fold change| ≥ 1 and *p*-value < 0.05). (**B**) Quantity statistics Venn diagram of DEGs among three sampling groups (|log2Fold change| ≥ 1 and *p*-value < 0.05).

**Figure 4 ijms-23-11621-f004:**
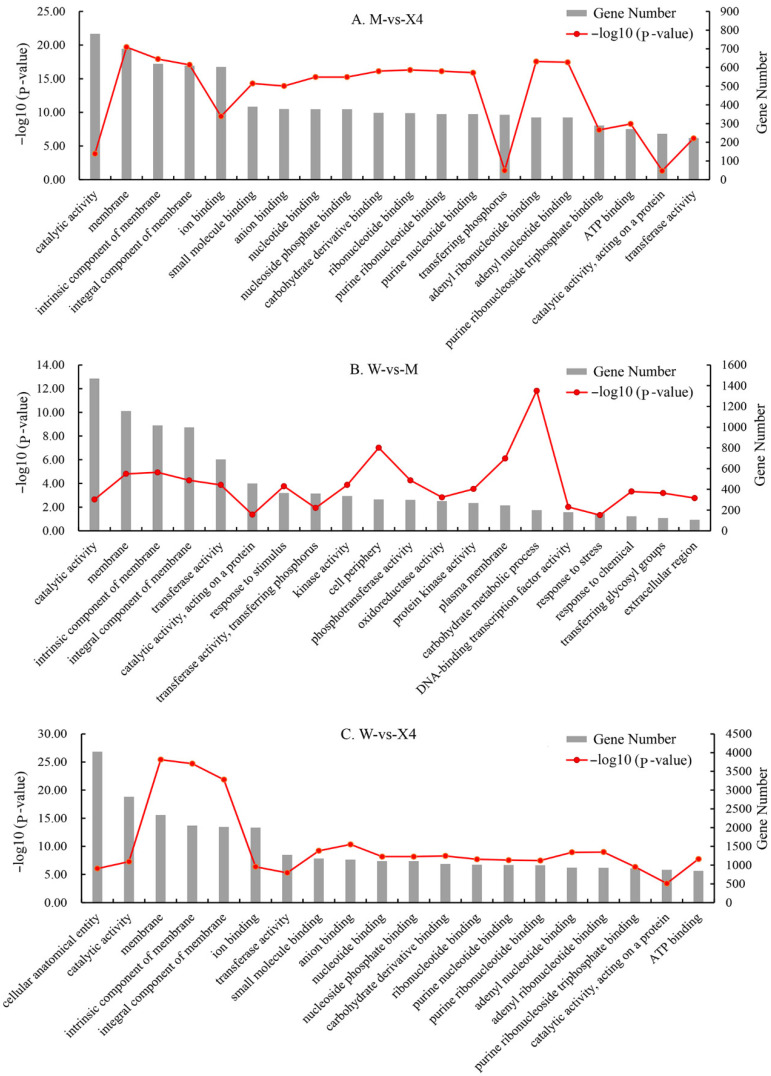
GO pathway analysis of differentially expressed genes. *X*-axis represents GO terms, left *Y*-axis represents *p*-value of specific GO term [expressed as −log10 (*p*-value)], right *Y*-axis represents number of DEGs of each GO pathway.

**Figure 5 ijms-23-11621-f005:**
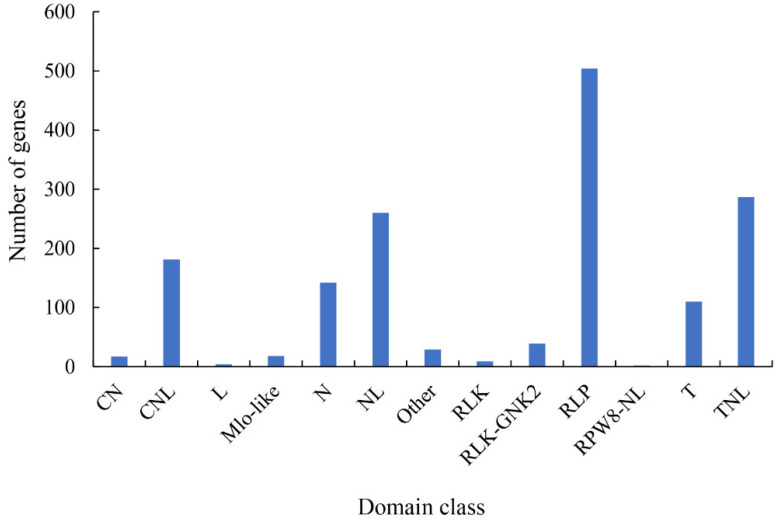
Domain class identification of DEGs annotated as pathogen-related genes (PRGs). Class descriptions available online (http://prgdb.crg.eu/wiki/Category:Classes, accessed on 3 June 2021).

**Figure 6 ijms-23-11621-f006:**
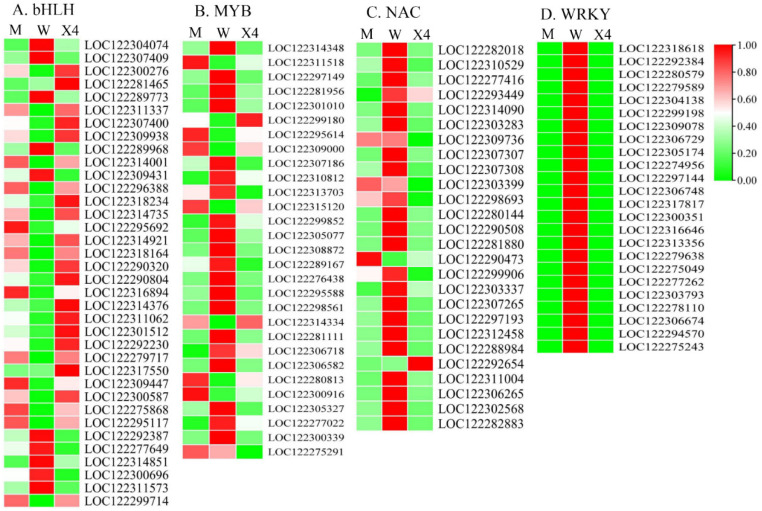
Heat map representing major transcription factor family members that were differentially expressed in fruits of three pecan cultivars: (**A**) bHLH; (**B**) MYB; (**C**) NAC; (**D**) WRKY. Colored boxes indicate genes with high (dark red) and low (dark green) expression.

**Figure 7 ijms-23-11621-f007:**
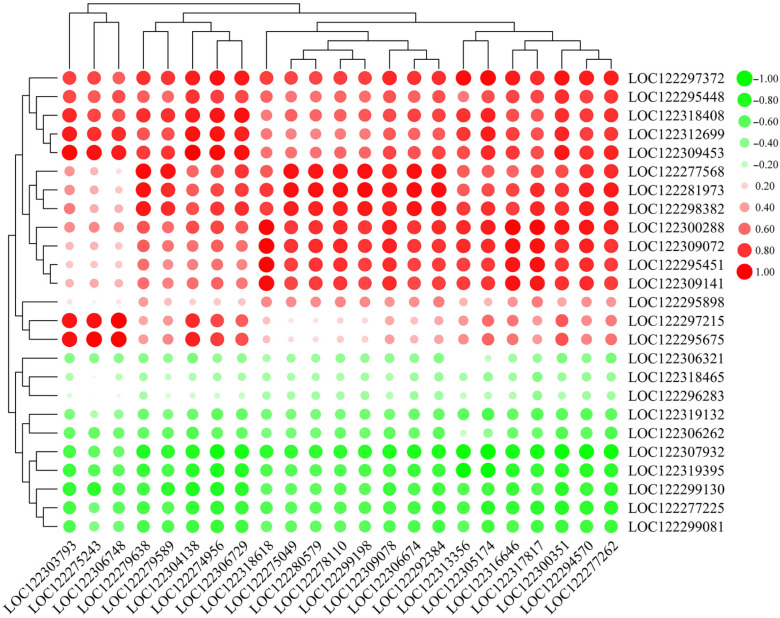
Correlation heat map of pathogen-related genes (PRGs) and WRKY TFs. Positive and negative correlations indicated by red and green circles, respectively; size of circle and intensity of color represent correlation strength. *X*-axis shows gene IDs of WRKY TFs, *Y*-axis shows gene IDs of PRGs.

**Figure 8 ijms-23-11621-f008:**
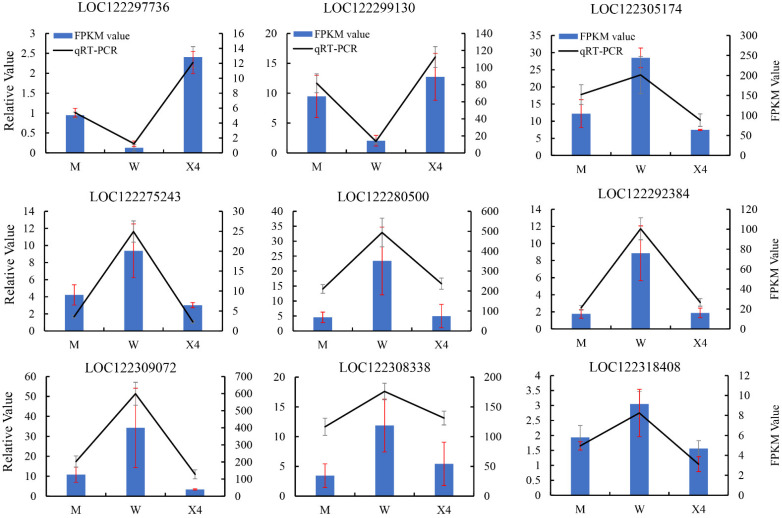
qRT-PCR validation of DEGs in three pecan fruit cultivars after FBS infection. Black line: qRT-PCR expression; blue bars: transcriptomic RNA-Seq analysis (FPKM). Primers and annotations of nine genes are listed in Appendix A.

**Table 1 ijms-23-11621-t001:** Statistical results of RNA-Seq.

Sample	Total Clean Reads (M)	Total Mapped Reads (%)	Unique Mapped Reads (%)	Clean Bases (Gb)	Clean Reads Q30 (%)
M_1	44.23	88.06	70.79	6.63	91.63
M_2	43.53	86.66	68.62	6.53	91.01
M_3	40.76	86.57	68.57	6.11	91.12
W_1	45.46	84.11	65.08	6.82	91.03
W_2	44.14	84.33	65.45	6.62	91.16
W_3	42.22	86.17	66.87	6.33	90.98
X4_1	44.18	85.25	66.21	6.63	90.81
X4_2	40.6	87.72	69.71	6.09	91.94
X4_3	45.35	86.76	68.64	6.8	91.95

**Table 2 ijms-23-11621-t002:** Significantly enriched pathways of DEGs among W, M, and X4 samples through KEGG enrichment analysis.

Pathway ID	Pathway	DEGs with Pathway Annotation (**log_2_ (Fold-Change**) > 1)			*p*-Value (<0.05)
		Total	Upregulated	Downregulated	
		**M vs. X4**			
ko04626	Plant–pathogen interaction	249	116	133	5.66 × 10^−16^
ko00195	Photosynthesis	29	28	1	1.61 × 10^−9^
ko04016	MAPK signaling pathway, plant	155	76	79	9.28 × 10^−8^
ko00500	Starch and sucrose metabolism	150	79	71	2.64 × 10^−7^
ko00051	Fructose and mannose metabolism	43	25	18	3.74 × 10^−7^
ko00196	Photosynthesis, antenna proteins	11	11	0	2.16 × 10^−5^
ko00965	Betalain biosynthesis	12	12	0	2.80 × 10^−5^
ko00901	Indole alkaloid biosynthesis	13	13	0	3.02 × 10^−5^
ko04075	Plant hormone signal transduction	139	75	64	0.000291
ko00030	Pentose phosphate pathway	41	23	18	0.000543
ko00592	Alpha-linolenic acid metabolism	27	11	16	0.006199
ko00630	Glyoxylate and dicarboxylate metabolism	30	17	13	0.01525
ko01040	Biosynthesis of unsaturated fatty acids	15	5	10	0.039254
		**W vs. M**			
ko04016	MAPK signaling pathway, plant	236	60	176	1.32 × 10^−8^
ko04075	Plant hormone signal transduction	223	106	117	1.43 × 10^−5^
ko01040	Biosynthesis of unsaturated fatty acids	24	10	14	0.011575
ko04626	Plant–pathogen interaction	290	84	206	0.011575
		**W vs. X4**			
ko04016	MAPK signaling pathway, plant	489	162	327	4.37 × 10^−24^
ko04626	Plant–pathogen interaction	651	238	413	2.04 × 10^−16^
ko00196	Photosynthesis, antenna proteins	23	23	0	6.00 × 10^−10^
ko04075	Plant hormone signal transduction	421	230	191	7.61 × 10^−9^
ko00195	Photosynthesis	48	46	2	1.55 × 10^−6^
ko00592	Alpha-linolenic acid metabolism	78	25	53	3.50 × 10^−6^
ko01040	Biosynthesis of unsaturated fatty acids	44	15	29	8.08 × 10^−5^
ko00564	Glycerophospholipid metabolism	107	60	47	0.001253
ko00051	Fructose and mannose metabolism	77	52	25	0.024548
ko00965	Betalain biosynthesis	16	16	0	0.024742

## Data Availability

All data are included in this published article. Raw RNA-Seq data generated in this study were deposited in NCBI SRA, available under accession number PRJNA824689.

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
