# Peer review of "Transcriptomic Analysis to Unravel Potential Pathways and Genes Involved in Pecan (Carya illinoinensis) Resistance to Pestalotiopsis microspora"

_ijms, 2022, doi:10.3390/ijms231911621_

Round 1

Reviewer 1 Report

Authors have studied the transcriptomic analysis of three different pecan cultivars that differ in their levels of resistance to FBS. This study has provided the RNA-seq data that could be helpful in understanding the pathogen resistance in Pecans.

Authors have attempted well in writing the manuscript. However, the manuscript needs an extensive grammatical corrections because it was really difficult to understand the manuscript. Moreover, authors have highly focused on the the susceptible variety but I would recommend them to include the details of the genes or TF's or pathways specific to the resistant variety as well. 

Please find the additional minor corrections and comments mentioned in the edited version of the manuscript. Please excuse the formatting of the document. Authors are advised to check the font size of the manuscript and also to italicize the organism name wherever applicable. 

Author Response

Review 1:

Authors have studied the transcriptomic analysis of three different pecan cultivars that differ in their levels of resistance to FBS. This study has provided the RNA-seq data that could be helpful in understanding the pathogen resistance in Pecans.

Comment: Authors have attempted well in writing the manuscript. However, the manuscript needs an extensive grammatical corrections because it was really difficult to understand the manuscript. Moreover, authors have highly focused on the susceptible variety but I would recommend them to include the details of the genes or TF's or pathways specific to the resistant variety as well.

Response: Thank you for your suggestions, the manuscript has undergone English language editing by MDPI (the figure below). The text has been checked for correct use of grammar and common technical terms. In the revised manuscript (results section), we have added the details of the genes/TFs/pathways specific to the resistant variety (Xinxuan-4).

Comment: Please find the additional minor corrections and comments mentioned in the edited version of the manuscript. Please excuse the formatting of the document. Authors are advised to check the font size of the manuscript and also to italicize the organism name wherever applicable.

Response: Thank you for your comments, the minor corrections and comments has been revised, we have also checked the font size or other format errors through the manuscript and it has been incorporated in the revised manuscript.

Reviewer 2 Report

The subject is interesting and worth studying .However the manuscript as it is written is not acceptable for publication. English needs to be improved significantly and it helps the reader understand the subject easily.

Please see a few mistakes as I mentioned below but there are many mistakes that can be found.

Line 27 -Disease resistance protein was performed qpcr, instead better revise the sentence validation etc

 Line 39-Remove Of, replace with a

Line  45 -Remove S from pathogens, It should be pathogen.

Line 51- Please revise what it means by first breeder ? its bit if confusion not clear

Line  54- Check PAMPs full form. It should be Pathogen-associated molecular pattern molecules (PAMPs).

Line 83- Remove for enhanced and keep to enhance.

Line 88-Remove plants. It should be plant-pathogen interactions.

Line 95– Rephrase this line. For example ,categorized into X4,M and W which may show low ,medium and high susceptibility to FBS infestation respectively.

Line -112-Keep one three. numerical or letter form. And in the same line remove to. It should be against infection.

Line 135- Remove Diagrams, write down diagrams because there is only one Venn diagram.

Line  154 - write down pathways.

Line 165- remove and. Write down there. The line should be mapped to the DEGs and ....

again remove that before the homologue and keep the comma after identified and start with the homologue.

Just reframe the sentence. if possible break it into two parts.

Line 322 - Correct electrophoresis stripes ? what it means by. Did you check RNA integrity value (RIN) if so mention.

Line 368- QRT-PCR please use proper terms. 

Author Response

Review 2:

Comment: The subject is interesting and worth studying. However the manuscript as it is written is not acceptable for publication. English needs to be improved significantly and it helps the reader understand the subject easily.

Response: Thank you for your suggestions, the manuscript has undergone English language editing by MDPI (the figure below). The text has been checked for correct use of grammar and common technical terms. Please see a few mistakes as I mentioned below but there are many mistakes that can be found.

Comment: Line 27 -Disease resistance protein was performed qpcr, instead better revise the sentence validation etc

Response: The suggested changes have been incorporated in the revised manuscript.

Comment: Line 39-Remove Of, replace with a

Response: The suggested changes have been incorporated in the revised manuscript.

Comment: Line 45 -Remove S from pathogens, It should be pathogen

Response: The suggested changes have been incorporated in the revised manuscript.

Comment: Line 51- Please revise what it means by first breeder ? its bit if confusion not clear

Response: it has been replaced with ‘signal’

Comment: Line 54- Check PAMPs full form. It should be Pathogen-associated molecular pattern molecules (PAMPs).

Response: The suggested changes have been incorporated in the revised manuscript.

Comment: Line 83- Remove for enhanced and keep to enhance.

Response: The suggested changes have been incorporated in the revised manuscript.

Comment: Line 88-Remove plants. It should be plant-pathogen interactions.

Response: The suggested changes have been incorporated in the revised manuscript.

Comment: Line 95– Rephrase this line. For example, categorized into X4, M and W which may show low, medium and high susceptibility to FBS infestation respectively.

Response: This part has been rephased.

Comment: Line -112-Keep one three. numerical or letter form. And in the same line remove to. It should be against infection.

Response: The suggested changes have been incorporated in the revised manuscript.

Comment: Line 135- Remove Diagrams, write down diagrams because there is only one Venn diagram.

Response: The suggested changes have been incorporated in the revised manuscript.

Comment: Line 154 - write down pathways.

Response: The suggested changes have been incorporated in the revised manuscript.

Comment: Line 165- remove and. Write down there. The line should be mapped to the DEGs and ....

again remove that before the homologue and keep the comma after identified and start with the homologue.

Just reframe the sentence. if possible break it into two parts.

Response: The suggested changes have been incorporated in the revised manuscript.

Comment: Line 322 - Correct electrophoresis stripes? what it means by. Did you check RNA integrity value (RIN) if so mention.

Response: Correct electrophoresis stripes means the clearly visible 5S, 18S, and 28S ribosomal RNA bands, it has been incorporated in the revised manuscript.

Comment: Line 368- QRT-PCR please use proper terms.

Response: The suggested changes have been incorporated in the revised manuscript.
